# Local and Systemic Inflammation in Finnish Dairy Cows with Digital Dermatitis

**DOI:** 10.3390/ani14030461

**Published:** 2024-01-31

**Authors:** Hertta Pirkkalainen, Aino Riihimäki, Taru Lienemann, Marjukka Anttila, Minna Kujala-Wirth, Päivi Rajala-Schultz, Heli Simojoki, Timo Soveri, Toomas Orro

**Affiliations:** 1Department of Production Animal Medicine, Faculty of Veterinary Medicine, University of Helsinki, Paroninkuja 20, 04920 Saarentaus, Finland; aino.riihimaki@helsinki.fi (A.R.); minna.kujala-wirth@helsinki.fi (M.K.-W.); paivi.rajala-schultz@helsinki.fi (P.R.-S.); heli.simojoki@helsinki.fi (H.S.); timo.soveri@helsinki.fi (T.S.); 2Animal Health Diagnostic Unit, Finnish Food Authority, Mustialankatu 3, 00790 Helsinki, Finland; taru.lienemann@ruokavirasto.fi (T.L.); marjukka.anttila@luukku.com (M.A.); 3Institute of Veterinary Medicine and Animal Sciences, Estonian University of Life Science, Kreutzwaldi 62, 51006 Tartu, Estonia; toomas.orro@emu.ee

**Keywords:** digital dermatitis, dairy cow, acute phase protein, cytokine, histopathology

## Abstract

**Simple Summary:**

The environment seems to first damage the skin allowing secondary bacteria to cause a local skin disease in bovines: digital dermatitis. Markers of inflammation, cytokines, are activated, but this does not initiate a systemic reaction and thus the body does not seem to be able to eliminate the disease.

**Abstract:**

Digital dermatitis is a disease of the digital skin and causes lameness and welfare problems in dairy cattle. This study assessed the local and systemic inflammatory responses of cows with different digital dermatitis lesions and compared macroscopical and histological findings. Cow feet (*n* = 104) were evaluated macroscopically and skin biopsies histologically. Serum samples were analyzed for acute phase proteins (serum amyloid A and haptoglobin) and pro-inflammatory cytokines (interleukin-1 beta, interleukin-6, and tumor necrosis factor-alpha). Cows with macroscopically graded active lesions (*p* = 0.028) and non-active lesions (*p* = 0.008) had higher interleukin-1 beta levels in their serum compared to healthy cows. Interleukin-1 beta serum concentrations were also higher (*p* = 0.042) when comparing lesions with necrosis to lesions without necrosis. There was no difference when other cytokine or acute phase protein concentrations in healthy cows were compared to those in cows with different digital dermatitis lesions. A novel histopathological grading was developed based on the chronicity of the lesions and presence of necrosis and ulceration. The presence and number of spirochetes were graded separately. In the most severe chronic lesions, there was marked epidermal hyperplasia and hyperkeratosis with necrosis, deep ulceration, and suppurative inflammation. Spirochetes were found only in samples from necrotic lesions. This study established that digital dermatitis activates proinflammatory cytokines. However, this did not initiate the release of acute phase proteins from the liver. A histopathological grading that takes into account the age and severity of the lesions and presence of spirochetes was developed to better understand the progression of the disease. It is proposed that necrosis of the skin is a result of ischemic necrosis following reduced blood flow in the dermal papillae due to pressure and shear stress caused by thickened epidermis, and that the spirochetes are secondary invaders following tissue necrosis.

## 1. Introduction

Bovine digital dermatitis (DD) is a hoof disorder located in the digital skin [1]. DD has been reported to cause systemic effects such as a decrease in milk yield and fertility problems [2,3]. The disease has been found worldwide [4] and there are multiple stages (M1–M4.1) [5,6]. M1, M2, and M4.1 lesions are considered active, and M3 and M4 lesions inactive [7]. Defining lesion stage by macroscopic evaluation can be difficult, and intra- and interobserver agreement has varied widely [8,9,10].

DD is thought to be a polymicrobial disease [11] and spirochetes from the genus Treponema have been considered the main causative agent [12,13,14]. Several studies have described the histopathological changes in bovine skin affected by DD. Döpfer et al. [6] described the histological changes in different M-stages. Thickening of the epidermis, superficial necrosis, hyperkeratosis, and infiltration of inflammatory cells are found in the bovine skin affected by DD [15,16,17]. Studies have shown that DD lesions extending deeper into the skin have a larger amount of and various Treponema species compared to superficial DD lesions [11,12,13,14,15,16,17,18].

Local and systemic inflammatory responses have the same goal: to destroy or capture infectious agents, remove damaged tissue, and repair the affected tissue [19,20]. Pro-inflammatory cytokines such as interleukin-1 beta (IL-β), interleukine-6 (IL-6), and tumor necrosis factor-alpha (TNF-α) are secreted locally at the site of injury [19,20]. In the liver they induce the production of acute phase proteins (APPs) such as serum amyloid A (SAA) and haptoglobin (Hp) [19,20]. Many different pathological conditions have been shown to elevate bovine serum SAA and Hp, i.e., mastitis [21,22], respiratory infection [23], and hoof disorders [24,25,26,27]. However, few studies have compared findings for local and systemic inflammatory responses.

In this study we assessed the local tissue damage and systemic inflammatory response of cows with DD and developed a histopathological grading system that more accurately describes DD lesions.

## 2. Materials and Methods

### 2.1. Study Herds

Data for this study were gathered in 2018. A more detailed description is available in Pirkkalainen and Riihimäki et al. [28]. The Natural Land Survey of Finland performed computerized random sampling of all Finnish free-stall dairy herds with at least 45 milking cows to select 250 dairy farms, stratified by the 15 regional Centers of Economic Development, Transport, and Environment (ELY Centres), and weighted by the dairy cow density in the region. The assumed DD M2 prevalence was 5% and to be able to estimate it with 5% precision, using 80% power, an Epitools sample size calculator was used [29]. The minimum herd size for the sampling was set at 45 dairy cows to capture mostly free-stall herds because Finland still has a large portion of small tie-stall herds. Since it was presumed that some farmers did not want to participate, the original sample size was larger. The final number of selected herds was 81. For sampling, 25 farms were chosen based on the availability of a trimming chute or a professional hoof trimmer. Three additional herds were chosen based on the presence of DD. Six animals that were euthanized due to non-lameness-related reasons from six different herds were chosen as healthy controls for histopathological evaluation of the digital skin.

### 2.2. Study Population and Sampling

The digital lesions on the hind legs were scored with the aid of a mirror (inspection mirror, Biltema) and a flashlight in a milking parlor or pen. The three veterinarians that performed the scoring had studied the M-scoring system together with the aid of photographs taken from the M-stage lesions. A more detailed description is given in a previous article by Pirkkalainen and Riihimäki et al. [28]. From each herd, three to six cows with DD lesions and one cow with healthy skin (*n* = 151) were chosen for more detailed evaluation (Figure 1). For macroscopic evaluation and sampling, the cows were placed in a trimming chute and the affected foot was lifted. A brief clinical examination was performed to exclude other diseases. We measured the body temperature and observed whether the animal had ocular or nasal discharge or an abnormal respiratory rate. The farmer was asked whether the cow had had any signs of mastitis, lameness, or other diseases prior to sampling. In addition, records concerning hoof trimming were evaluated. Cows with concurrent pathologies were not taken into the study.

For comparison of epidermal thickness, biopsies were also taken from six cows originating from six different herds. These cows were euthanized in the Production Animal Hospital of the University of Helsinki due to non-lameness-related reasons and samples were taken postmortem. Those cows were taken as healthy controls as they did not have any DD lesions. Eventually, our study comprised 60 Finnish Holstein cows, 44 Finnish Ayrshire cows (*n* = 104) with serum and biopsy samples, and an additional six control cows on which only histological evaluation was performed (Figure 1).

A consensus of diagnosis was obtained by evaluating photos of the lesions together among the three researchers who visited farms. The other two researchers did not have any information on the cow or lesions prior to evaluation. Macroscopic lesions were classified to different M-stages described by Döpfer et al. [6], modified by Berry et al. [5]. M1 has been described as an early-stage ulcerative lesion (0–2 cm in diameter) that may develop into a M2 lesion. M2 is an ulcerative stage with a diameter >2 cm and is often painful on palpation. M3 is a lesion covered by a scab and healing. M4 is considered a chronic stage characterized by dyskeratosis or a corn-like proliferation. M4.1 is a chronic lesion with a small area of new ulceration [5,6]. Skin without any DD lesions is classified as M0 [5,6]. Lesions were also categorized active (M1, M2, and M4.1) and inactive (M3, M4) [7]. All cows also had a diagnosis of the other hind leg. In cows with DD lesions, the foot with the most severe lesion was chosen for sampling and thus the final analysis was performed on an animal-level.

Prior to sampling, 10 mL of procaine (Procamidor vet. 20 mg/mL; VetViva Richter GmbH, Durisolstrasse 14, 4600 Wels, Austria) was injected under the skin a few centimeters below the dewclaws. Biopsies were taken with a sterile 6 mm biopsy punch (Kruuse, Odense, Denmark) with the help of sterile surgical forceps. Two adjacent biopsies were taken from the center of each lesion with the same biopsy punch. The first tissue sample was placed in RNA stabilization solution (RNAlater^®^, Thermo Fisher, Waltham, MA, USA) and the second in a vial with 7 mL of formalin (Formalin 10%, Reagena Oy Ltd., Toijala, Finland). Only one foot per cow was accepted for this study. Blood was collected into serum tubes (BDVacutainer^®^, Franklin Lakes, NJ, USA) either from jugular, coccygeal, or subcutaneous abdominal veins. We sprayed the biopsy site with tetracycline spray (Engemycin 25 mg/mL, MSD Animal Health/Intervet, Aprilia, Italy) and made a hoof wrap with non-sterile gauze (Gauze swabs, Mölnlycke Health Care AB, Gothenburg, Sweden), non-sterile elastic fixation bandage, Elastic Fixation Bandage Fix Premium, 10 cm × 4 m, OneMed Group Oy, Helsinki, Finland), and a flexible bandage (Vet-Flex Flexible Bandage, 10 cm × 4.5 m, Kruuse, Odense, Denmark). We filled in a form about the herd-, animal-, and biopsy-related information and marked the sample vials with herd-, animal-, and biopsy-related information.

Samples were packed and sent via parcel service (Oy Matkahuolto ab, Helsinki, Finland) within 24 h to the laboratory of the Production Animal Hospital of the University of Helsinki. The serum was separated by centrifugation (2500 rpm) and divided into two aliquots. Both serum and RNAlater tissue samples were frozen at −20 °C. Formalin samples were kept at room temperature until the moment of histopathological evaluation. When all the skin biopsies were collected, they were sent to a histopathology laboratory for tissue processing and slide preparation (Finnish Food Authority laboratory in Helsinki).

### 2.3. Analysis of Blood Samples

Markers of acute inflammation, the APPs SAA and Hp and the pro-inflammatory cytokines IL-1β, IL-6, and TNF-α concentrations, were measured from serum samples at the Institute of Veterinary Medicine of the Estonian University of Life Sciences. As sample hemolysis can influence results, the hemolysis of samples (*n* = 104) was visually estimated and samples categorized as follows: no hemolysis (*n* = 95), mild hemolysis (*n* = 3), moderate hemolysis (*n* = 2), and severe hemolysis (*n* = 4). 

A commercial sandwich enzyme-linked immunosorbent assay (ELISA) kit (Phase Serum Amyloid A Assay (SAA) Multispecies, Tridelta Development Ltd., Maynooth, Co. Kildare, Ireland) was used to determine SAA concentrations in serum samples. The manufacturer’s instructions for cattle were followed in the analysis, calibration, and dilution protocols. The initial sample dilution used was 1:500. A calibration curve was made for analyzing the serum samples (high value of 300 mg/L). If the results of the samples exceeded the measuring range, the samples were diluted (1:2000) and reanalyzed. Intra- and inter-assay CV% were <13% and the lower detection limit of the analytical procedure was 0.3 mg/L.

Serum haptoglobin analyses were performed according to the method suggested by Makimura et al. with a modification [30] using tetramethylbenzidine (0.06 mg/mL) as a substrate instead of o-dianisidine. A pooled and lyophilized bovine acute phase serum aliquot was used to create a standard curve [30]. The calibration of a standard curve was achieved by using a bovine serum sample of known Hp concentration, which was provided by the European Commission Concerted Action Project (number QLK5-CT-1999-0153). The calibration curve ranged from 60 to 960 mg/L. Samples with higher results were re-assayed after a 1:5 dilution with 0.9% NaCl solution. The detection limit of the analytical procedure was 60 mg/L and intra- and inter-assay CV% were <12%.

Pro-inflammatory cytokine concentrations were analyzed using bovine IL-1β, IL-6, and TNF-α ELISA kits (Cusabio Biotech, Wuhan, China) according to the manufacturer’s instructions. The detection limits for IL-1β, IL-6, and TNF-α were 15.6 ng/L, 2.5 ng/L, and 50.0 ng/L, respectively. The inter- and intra-assay CV% of all cytokine detection methods were under 15%.

### 2.4. Histopathological Evaluation

Histopathological examination was performed at the Finnish Food Authority laboratory in Helsinki. The samples fixed in 10% neutral-buffered formalin were cut into 3 mm sections, routinely processed, embedded in paraffin, cut into 4 µm sections, and stained with hematoxylin and eosin (HE) and Warthin–Starry (WS). The lesions were evaluated, and the findings were graded and placed in seven groups (Table 1). The grading of the histological changes was based on the age and severity of the lesions. The evaluation of the age of the lesion was based on the thickness of the epidermis (Table 2) and the severity was based on the depth of the necrosis. The thickness of the epidermis was measured without the keratinized layer. The number of spirochetes was evaluated from samples stained with WS silver staining using a semi-quantitative scale: 0 = no spirochetes; 1 = low number; 2 = moderate number; 3 = large number of spirochetes.

The location of the silver-positive spiral bacteria was evaluated, and the findings were noted for bacteria on the surface of the skin within the keratin layer (S), bacteria within a zone of necrotic epidermis (NZ), and for bacteria multifocally in the necrotic epidermis around the necrotic dermal papillae (MFN). In addition, the severity of the chronic dermal inflammation was graded (Table 3). The histopathological examination was performed without knowledge of the macroscopic lesion categorization.

### 2.5. Statistical Analysis

Statistical analyses were performed using data from 104 blood and histopathological samples (Figure 1). For evaluating associations between serum concentrations of APPs (SAA and Hp) and pro-inflammatory cytokines (IL-1β, IL-6, and TNF-α) with DD lesions and the histological grade of DD lesions, generalized linear models (glm) with gaussian dependent variable distribution were used. Pro-inflammatory cytokines and APPs were logarithmically transformed to obtain a normal distribution for these outcome variables. As sample hemolysis can affect blood measurements, four levels of a categorical variable (no hemolysis, slight hemolysis, moderate hemolysis, or severe hemolysis) were included in all models. For controlling a possible farm effect, generalized linear mixed models (glmm) with farm as the random intercept variable were used. As farm random intercepts were not significant in any models, glm models were used for the final analysis. To evaluate blood analyte associations with DD lesions, cow macroscopical DD lesion was included as a categorical variable at five levels (M0, M1, M2, M3/M4, and M4.1). Non-active DD lesions (M3/M4; *n* = 29) were combined because only one cow had an M3 DD lesion. To evaluate blood analytes’ associations with active or chronic DD lesions, a three-level categorical variable (M0 = healthy, M1, M2, and M4.1 = active lesion, and M3/M4 = chronic lesion) was used. A Wald test was used to evaluate a more than two-level categorical variable overall significance in the model. Bonferroni-corrected *p*-values were used for pairwise comparisons between the levels of those 5- and 3-level categorical variables. To evaluate serum APP and pro-inflammatory cytokine associations with lesions’ histological grade, the DD lesion histological grade was included as a seven-level categorical explanatory variable. In those models, Bonferroni-corrected *p*-values were used for pairwise comparison. Two-level categorical variables were constructed to evaluate differences between cows with similar histological grades, e.g., comparing histological grades with necrosis (2, 3, 5, and 6) to histological grades without necrosis (0, 1, and 4) or severe necrosis (5 and 6) compared to the least severe histological changes (0 and 1).

Assumptions for all models were checked using normality and scatter plots of the model residuals. The significance level was set at *p* ≤ 0.05 and tendency level as *p* = 0.051–0.099. Statistical analyses were performed using Stata 14.2 (StataCorp LP, College Station, TX, USA). Microsoft Excel for Mac version 16.35 (Microsoft, Redmond, Washington, DC, USA) was used for data management.

## 3. Results

Macroscopically, 35 M0, seven M1, 19 M2, one M3, 28 M4, and 14 M4.1 lesions formed a total of 104 diagnoses (Table 2). All six additional biopsy samples were macroscopically diagnosed as M0. However, these additional samples were not included in the statistical analysis. Analyses concerning APPs and cytokines were performed on data from the 104 cows.

### 3.1. Systemic and Local Inflammatory Response to DD Lesions

IL-1β serum concentrations were higher in DD M3/M4 lesions (*p* = 0.016) compared to healthy cows (overall Wald test for DD lesions *p* = 0.023). In addition, cows with DD M4.1 lesions had higher IL-1β and TNF-α serum levels than healthy cows (tendency, *p* = 0.060 and *p* = 0.092, respectively). IL-6 and APP (SAA and Hp) concentrations did not differ between healthy cows and those with different DD lesions.

A comparison of cytokine serum concentrations in healthy cows to cows with active and non-active DD lesions is shown in Figure 2. Cows with macroscopically graded active DD lesions (*p* = 0.028) and non-active DD lesions (*p* = 0.008) had higher IL-1β levels in their serum compared to healthy cows (overall Wald test for active and non-active DD lesions *p* = 0.008). TNF-α was higher in non-active DD lesion cows compared to healthy cows (*p* = 0.048; overall Wald test *p* = 0.072) and IL-6 was higher in cows with non-active lesions compared to healthy cows (tendency, *p* = 0.068). There was no difference when APP (SAA and Hp) concentrations in healthy cows were compared to active and non-active DD lesion cows. 

IL-1β serum concentrations were higher (coefficient ± SE: 0.779 ± 0.384 (log)ng/L; *p* = 0.042) when comparing histological grades with necrosis (2, 3, 5, 6, *n* = 65) to histological grades without necrosis (0, 1, 4; *n* = 39). When cows with the most severe necrosis (grades 5, 6; *n* = 51) were compared to the least severe histological changes (grades 0, 1; *n* = 35), IL-1β was higher (coefficient ± SE: 0.851 ± 0.406 (log)ng/L; *p* = 0.036). When the most severe grades (5, 6; *n* = 51) were compared to all other histological grades (0, 1, 2, 3, 4, *n* = 53), IL-1β was higher (coefficient ± SE: 0.829 ± 0.374 (log)ng/L; *p* = 0.027). When comparing the most severe grades (5, 6; *n* = 51) to changes without necrosis (0, 1, 4; *n* = 39), IL-1β was significantly higher (coefficient ± SE: 0.913± 0.386 (log)ng/L; *p* = 0.018). Histopathological grading is explained in Table 1.

### 3.2. Histopathological Changes

The grading of the histopathological lesions is shown in Table 2 and illustrative histological pictures of several different grades are shown in Figure 3, Figure 4, Figure 5, Figure 6 and Figure 7.

In 30 biopsies from M0 lesions, the skin was normal or had mild changes. Five samples that were visually graded as healthy had distinct thickening of the skin. All lesions graded as M2 or M4.1 had skin necrosis. In the control skin samples from six slaughtered healthy animals, the thickness of the epidermis was below 0.8 mm regardless of the age of the animal (average age 7.5 years; range 2–10 years).

In the samples with epidermal hyperplasia, the surface of the hyperplastic epidermis was usually even but at the lower margin the epidermis formed deep projections into the dermis, resulting in the formation of dermal papillae between the epidermal projections (Figure 3b). Within the thickened keratin layer there were vertical fissures and horizontal splitting in some samples.

The necrosis of the epidermis and dermis was multifocal, locally extensive, or diffuse. There was acute necrosis of the dermal papillae with a variable degree of neutrophilic reaction. In the cases with multifocal necrosis, there was acute necrosis of the dermal papillae with necrosis of the epidermis surrounding the papillae. In cases with diffuse or locally extensive necrosis, there was necrosis of the dermal papillae and necrosis of the intervening epidermis. In some cases, the loss of necrotic epidermis and keratin resulted in the formation of filamentous naked dermal papillae. In some samples there were horn-like structures protruding from the skin due to the multifocal loss of surrounding keratin. In some cases, the surface of the epidermis was papillary and there was no keratin layer due to multifocal loss of the surrounding epidermis and sloughing off of the keratin layer. In the most severe cases, there was deep focal ulceration that extended to the nonpapillary dermis with marked acute purulent inflammation concurrent with the more chronic lymphoplasmacytic inflammation. The dermal inflammation consisted of lymphocytes and plasma cells in samples with no necrosis or ulceration.

There were no spirochetes in the samples that were normal or had mild to marked epidermal hyperplasia without necrosis, i.e., in the samples graded 0, 1, and 4 (Table 3). Spirochetes were found only in samples that had epidermal and dermal necrosis. No spirochetes were present within the keratin layer if there was no tissue necrosis. Spirochetes were present in all but seven of the 65 samples with necrosis, i.e., grades 2, 3, 5, and 6. The largest number of spirochetes was in the samples with a wide zone of necrotic epidermis; in 20 out of 22 samples with grade 3, spirochete numbers had a wide zone of epidermal necrosis. In a few cases spirochetes were found within the necrotic dermal papillae. 

## 4. Discussion

### 4.1. Acute Phase Proteins and Cytokines

Few studies have investigated the systemic inflammatory response in DD. In our earlier study [27], we reported elevated IL-6 levels but did not find elevated SAA or Hp levels in bovine serum when comparing active DD lesion cows and DD-free control cows. Ilievska et al. [25] reported higher SAA and Hp levels in the venous blood of DD cows when compared to healthy controls. However, the number of animals in both studies was relatively small. In our study, the level of proinflammatory cytokines was higher among DD cows when compared to cows without DD lesions. However, we did not record increased APP levels in cows with DD compared to DD-free cows. It is noteworthy that we collected blood samples only once, and we do not know how long different lesions had existed. Interestingly, when we compared histopathological grades to APPs and cytokines, we only found a significant increase in IL-1β when comparing necrotic lesions to lesions without necrosis. This could be due to the small number of lesions in certain categories (2, 3). We suggest that when comparing macroscopical grading to histological grading, the previous is correlated better with the increase in cytokines.

In general, wound healing can be divided into different phases as for hemostasis, inflammation, proliferation, and remodeling [31]. Cytokines play an important role in the wound healing process, and we recorded elevated IL-1β levels from cows with necrotic DD lesions. In an acute injury, prestored IL-1β is released from keratinocytes, signaling other cells around the damaged area [32]. IL-1β is also known to attract neutrophils to the injury site to remove bacteria [33]. Additionally, growth factors play a key role in the complex wound healing process. In DD lesions, necrosis can lead to higher levels of IL-1 release and production, which then can be detected in the serum of cows [27]. However, this does not necessarily mean that their effect is sufficient to initiate the production of APPs in the liver. For example, sole ulcer—another hoof disorder with a different etiology—can elevate APPs, but the tissue damage in this condition is more severe compared to DD [27]. We could hypothesize that the bovine body is not able to eliminate DD from the skin because there is no efficient systemic reaction. Krull et al. [34] stated that the median time for DD lesion development was 105 days (range = 38–315 d). They also suggested that bacterial communities associated with DD change significantly as the lesions develop and that these bacterial communities are closely tied to the morphological stage of lesion development.

A few studies have highlighted the local inflammatory response in cows with DD [35,36,37]. Scholey et al. [35] studied the host pathogenic pathways. The local immune reaction to the bacterial infection present in DD lesions was very mild. Zuerner et al. [36] suggested that the innate immune and wound repair functions of bovine macrophages exposed to treponeme cellular components were impaired. This enabled bacteria to resist elimination and induce lesion formation [36]. Evans et al. [38] reported the upregulation of genes encoding several inflammatory mediators, such as TNF-α, in fibroblasts, but not in keratinocytes. These findings also could explain why we do not see a systemic inflammatory pathway in cows with DD. 

In addition to immunity, the environment must play an important role in such lesion development. Treponemas might be inhibitors of the healing process.

### 4.2. Histopathology

The thickness of epidermis did not increase with the age of the animal and thus appeared to be affected only by the environmental factors in healthy animals. The layer of keratinized cells was not included in the measurement because its thickness is affected by several factors, e.g., splitting, flaking, and erosion. The epidermis thickens by hyperplasia because of chronic inflammation and/or chronic irritation by noxious compounds, wetness, hard surfaces, etc. and it is usually associated with hyperkeratosis.

The necrosis of the dermal papillae appears to precede the necrosis of the epidermis and it is probably ischemic. The thickened epidermis causes pressure and shear stress on the delicate dermal papillae, which impairs the blood flow and causes hypoxia, leading to ischemic necrosis followed by erosion and ulceration. The fissures and splitting in the thickened keratin are likely caused by pressure and shearing.

Rasmussen et al. [39] investigated DD lesions from Danish and Norwegian herds. Their findings were similar to ours, although their lesions were not categorized as acute. They also suggested that the first phase of the disease development is non-infectious (e.g., external noxious) induced skin lesions, which leads to keratinization defects and secondary invasion of bacteria. According to our findings, the spirochetes appear to be present only in the necrotic skin. Thus, we hypothesize that they are secondary invaders. Long-standing irritation and moisture caused by a suboptimal environment causes thickening of the epidermis and hyperkeratosis and may alter the physical resistance and protective characteristics of the skin. The feet were found to be dirty in Finnish free-stalls [28]. This indicates that there are problems with manure removal. The environment in free-stall barns is not always optimal and there often are problems with manure removal, air conditioning, and ammonia. 

Our histological descriptions were similar to those of previous studies [6,14,39,40,41]. However, other studies used a different categorization of histological samples. With our novel grading, histopathologic lesion grades did not differ greatly among M2, M4, and M4.1 lesions. However, the M4 and M4.1 lesions tended to have higher grades in histopathological evaluation, indicating a long-lasting skin irritation. In a recent Finnish study, the within-herd prevalence of active lesions was 4.6%, which is quite low [28]. However, the mean apparent within-herd prevalence of any DD-like lesions was 30.9% (range 2.5–82.6%), which means the overall presence of DD is quite high in the herds. 

We can conclude that the clinical picture for DD in Finland is reasonably mild. There has not been much animal trade from other European countries to Finland in the last few decades [42]. In the histological evaluation, none of the grade 0, 1, and 4 lesions supported spirochetes. All but seven samples in the more severe lesion categories, i.e., 2, 3, 5, and 6, had spirochetes within the epidermis, only in the necrotic areas, thus indicating secondary invasion. There were seven samples with marked necrosis but no spirochetes. In addition to legislation, animal trade has been guided by an industry-based organization, Animal Health ETT, and the risk of infectious diseases is considered in imports [43]. As animal trade has been identified as a risk factor for DD [44], it can be hypothesized that we might not have imported that many virulent Treponemes to Finland. Based on our histological evaluations, we isolated spirochetes from our DD lesions. However, further bacteriological analyses are needed to gain knowledge about the bacteria involved.

Skin biopsies from samples that were macroscopically graded as M0 were histopathologically mainly scored as 0 or 1. However, two samples were graded as 4 and three samples as 5. As spirochetes were not found within the skin in histological categories 0, 1, or 4, we can conclude from the data that visual inspection can differentiate whether DD is present or not. High agreement on dichotomous DD evaluation has been determined in other studies, however they did not have a histological result as a reference [10].

### 4.3. Pain

Active DD lesions can be painful to the animal, resulting in posture and gait changes [45,46,47]. Shaking and lifting of feet, toe-down walking, and reluctance to walk are typical clinical signs of DD [48]. Severe lesions of DD and interdigital dermatitis/heel horn erosion (IDHE) have been associated with a significantly higher proportion of lame cows. The proportion of animals with disturbed locomotion increased from 16% to 40% as the severity of DD increased and from 17% to 30% with increasing severity of IDHE lesions [49].

Pain may affect milk production and fertility. A decrease in milk yield [2,50] and lower fertility [51] have been reported for cows with DD lesions. IL-1β, IL-6, and TNF-α are part of the up-regulation of inflammatory reactions [51]. They are also involved in the process of pathological pain [52]. As described earlier, there are indications of ischemic necrosis within the skin. We found severe thickening of the skin in most DD lesions. As discussed earlier, necrosis of the dermal papillae appears to precede the necrosis of the epidermis due to ischemia, resulting from excessive pressure caused by the thickened skin. Heel fissures are known to be painful in humans [53] and can even be considered a debilitating problem. Longhurst and Steele [54] explain that, with increased pressure heel splits become deeper, involving the dermis so that they begin to bleed and result in pain on weight-bearing activities. In many of the DD lesions we saw epidermal thickening, cracks within epidermis, necrosis of epidermis, and even complete loss of epidermis resulting in naked dermal papillae. We can hypothesize that this result is painful and can cause systemic effects.

### 4.4. Study Limitations

There were limitations to our study. The total number of cows examined was low (*n* = 104) and there were only 1–35 DD lesions per M-stage. Furthermore, we do not know to which type of genetic susceptibility of DD the examined cows belonged. Capion et al. [55] divided cows into three different susceptibility categories: consistently healthy cow (1); intermittently infected cow (2); and consistently infected cow (3). This aspect might have influenced the concentration of pro-inflammatory cytokines in the blood. Although we performed a brief clinical examination of cows, we might not have been able to rule out other metabolic diseases. However, cows with no DD lesions were also examined and sampled the same way. Some of the cows had another DD lesion in the other hind leg. This might have affected cytokine and APP concentrations in the bovine serum. However, we took this into account in the statistical analysis.

It would be interesting to repeat this type of study in a country where DD is more prevalent than it is in Finland. In addition, microbiome studies would shed light on the bacterial composition of different DD lesions and could be compared to healthy skin. 

## 5. Conclusions

This study established that DD activates proinflammatory cytokines. Necrosis in DD lesions can lead to higher levels of IL-1β release and production, which then can be detected in the serum of cows. However, this does not cause the release of acute phase proteins from the liver.

We propose a novel histopathologic grading based on the age and severity of the lesions and presence of necrosis that was developed to understand the progression of DD better. We propose that epidermal necrosis is a result of ischemic necrosis of the dermal papillae caused by pressure due to decreased pliability of the thickened skin and shear stress and that the spirochetes are secondary invaders in the necrotic skin.

## Figures and Tables

**Figure 1 animals-14-00461-f001:**
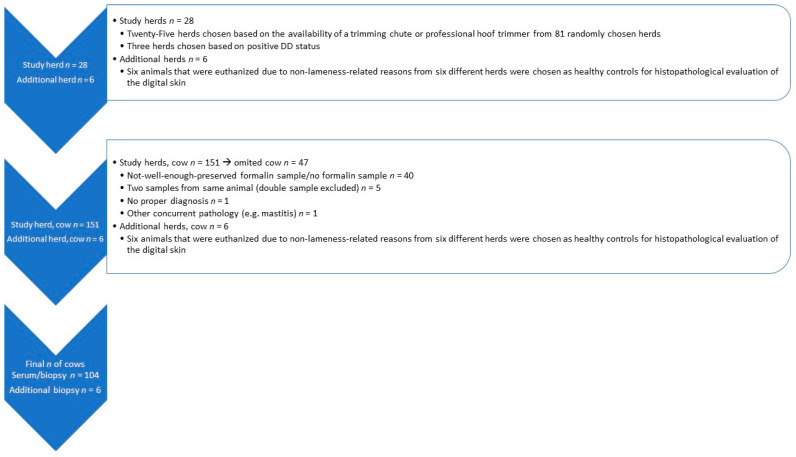
Flow chart for eligible serum samples and biopsies from digital skin on Finnish dairy cows with reasons for exclusions.

**Figure 2 animals-14-00461-f002:**
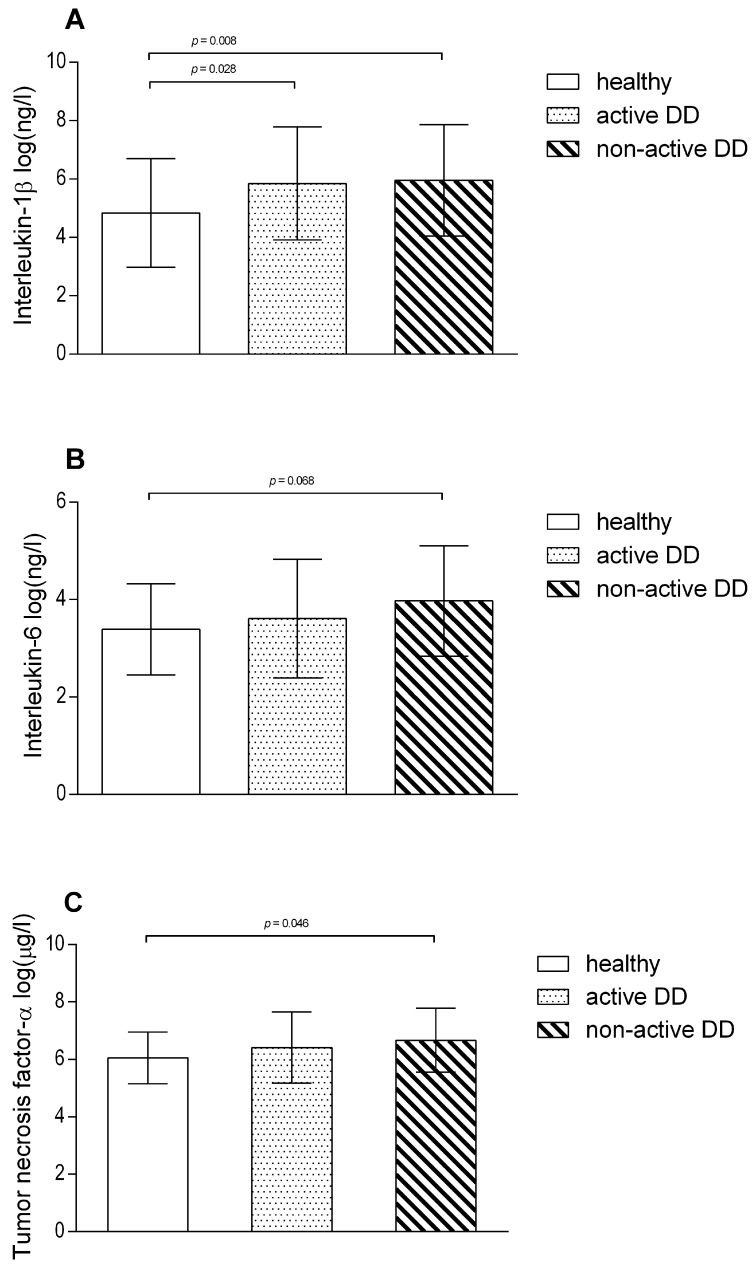
Interleukin-1β (**A**), interleukin-6 (**B**), and tumor necrosis factor-α (**C**) mean serum concentrations in logarithmic scale (±SD) in healthy cows (*n* = 35) and cows with active (M1, M2, and M4.1; *n* = 40) or non-active (M3/M4; *n* = 29) digital dermatitis (DD) lesions. Bonferroni-corrected *p*-values are presented.

**Figure 3 animals-14-00461-f003:**
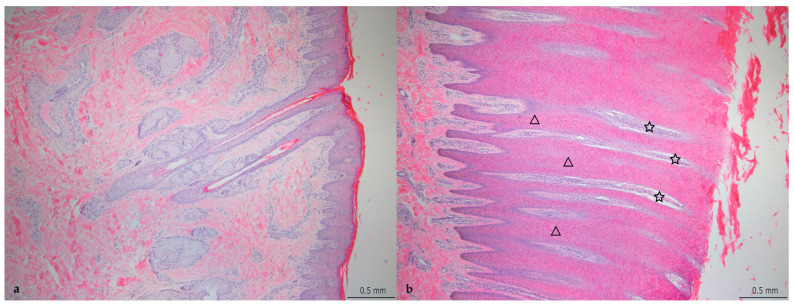
(**a**) Normal skin (histopathological grade 0, DD M0). The black bar is 0.5 mm in length (HE stain, ×4). (**b**) Chronic lesion of digital dermatitis without necrosis or inflammation (histopathological grade 4, DD M4). Marked increase in the epidermal thickness by hyperplasia. Some of the dermal papillae are marked with a star and some of the deep epidermal projections are marked with a triangle. The black bar is 0.5 mm in length (HE stain, ×4).

**Figure 4 animals-14-00461-f004:**
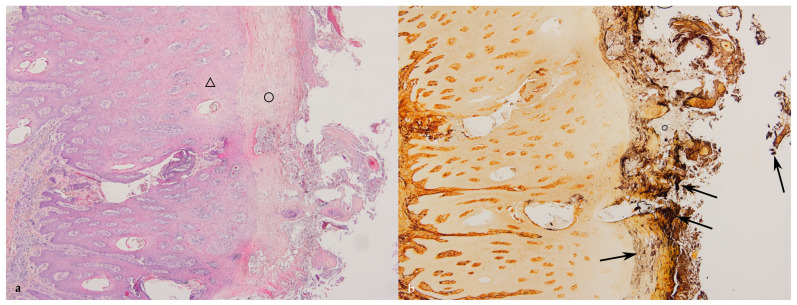
(**a**) Digital dermatitis, histopathological grade 5 (DD M4). Acute necrosis of the superficial part of the epidermis (HE stain, ×4). (**b**) Digital dermatitis, histopathological grade 5 (DD M4), same section as in (**a**). Spirochetes (stained black) are present on the surface of the epidermis and within the necrotic epidermis (black arrows). Viable epidermis is marked with a triangle and necrotic epidermis is marked with a circle (WS stain, ×4).

**Figure 5 animals-14-00461-f005:**
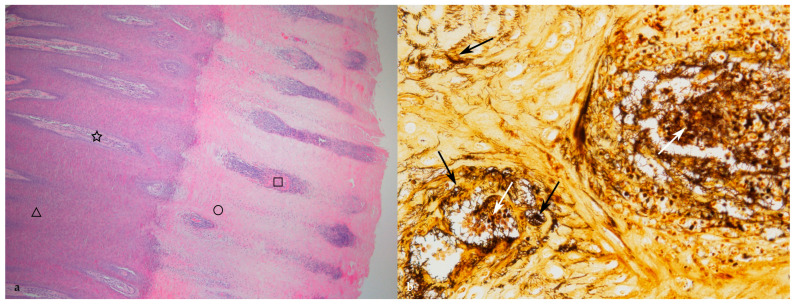
(**a**) Digital dermatitis, histopathological grade 5 (DD M4). Acute, locally diffuse necrosis of the superficial part of dermal papillae and the intervening epidermis. Viable epidermis is marked with a triangle, viable dermal papillae with a star, necrotic epidermis with a circle, and necrotic dermal papilla with a square (HE stain, ×4). (**b**) Digital dermatitis, histopathological grade 5 (DD M4.1). Spirochetes (stained black by silver stain) are present within the necrotic epidermis (black arrows) and in the dermal papillae (white arrows) (WS stain, ×40).

**Figure 6 animals-14-00461-f006:**
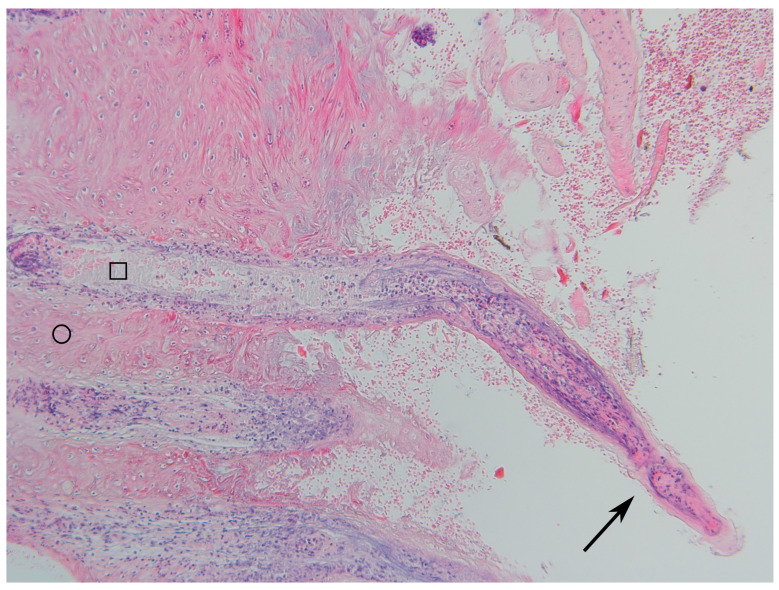
Digital dermatitis, histopathological grade 5 (DD M4.1). Naked, necrotic dermal papillae on the surface of the skin (black arrow) as a result of sloughing off of the necrotic epidermis. Necrotic epidermis is marked with a circle, and necrotic dermal papilla with a square (HE stain, ×10).

**Figure 7 animals-14-00461-f007:**
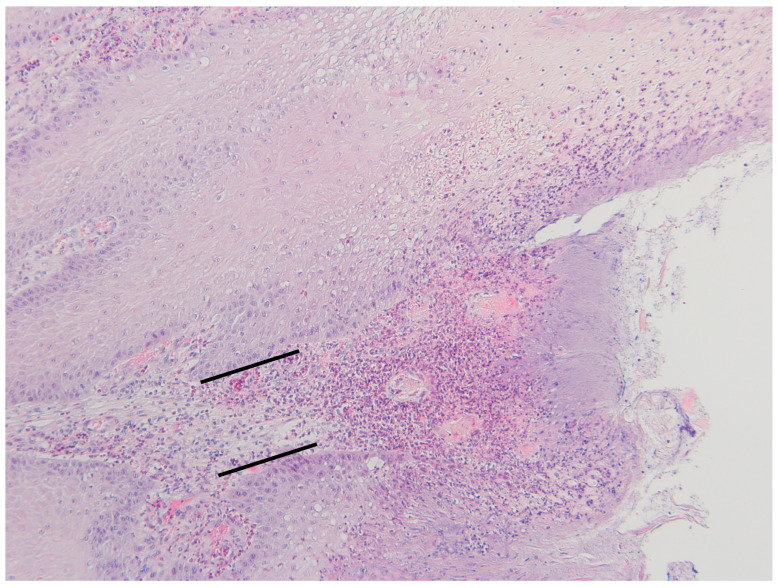
Digital dermatitis, histopathological grade 6 (DD M2). Acute ulceration of the skin extending deep into the epidermis and dermis with acute neutrophilic inflammation (area between black lines) (HE stain, ×10).

**Table 1 animals-14-00461-t001:** Grading of the histopathological changes and thickness of epidermis of skin biopsies from the hind feet of Finnish dairy cows with and without DD lesions.

Grade	Description	Thickness of Epidermis (mm)
0	normal	0.3–0.8
1	mild to moderate hyperplasia and ortokeratotic hyperkeratosis	1–1.5
2	low moderate hyperplasia and ortokeratotic hyperkeratosis with necrosis and erosion	1.5–2
3	high moderate hyperplasia and ortokeratotic hyperkeratosis with necrosis and ulceration extending to the nonpapillar dermis	2–2.4
4	marked hyperplasia and (para- and ortokeratotic) hyperkeratosis with some fissures in the thick layer of keratin	2.5–4.5
5	marked hyperplasia and (para- and ortokeratotic) hyperkeratosis with necrosis and erosion	4–4.5
6	marked hyperplasia and (para- and ortokeratotic) hyperkeratosis with necrosis and ulceration extending to the nonpapillar dermis	4–4.5

**Table 2 animals-14-00461-t002:** Number of cows with different histopathological grades according to macroscopic digital dermatitis (DD) lesions (*n* = 104).

DD Lesion	Histopathological Grade ^1^							Total (n)
	0	1	2	3	4	5	6	
M0	19	11	0	0	2	3	0	35
M1	2	2	1	0	1	1	0	7
M2	0	0	0	6	0	7	6	19
M3	0	0	0	0	0	1	0	1
M4	0	1	2	0	1	22	2	28
M4.1	0	0	5	0	0	8	1	14
Total (n)	21	14	8	6	4	42	9	104

^1^ Histopathological grade definitions are given in Table 1.

**Table 3 animals-14-00461-t003:** Number of spirochetes in different histopathological digital dermatitis (DD) lesion grades (semi-quantitative scale). The location of the silver-positive spiral bacteria was evaluated, and the findings were noted as bacteria within a zone of necrotic epidermis (NZ) and bacteria multifocally in the necrotic epidermis around the necrotic dermal papillae (MFN).

Grade ^1^	0 (No Spirochetes)	1 (Low No. of Spirochetes)	2 (Moderate No. of Spirochetes)	3 (Large No. of Spirochetes)
0	21	0	0	0
1	14	0	0	0
2	3	2 (NZ, MFN)	1 (NZ)	2 (NZ)
3	0	0	3 (NZ)	3 (NZ)
4	4	0	0	0
5	0	14 (1NZ, 13 MFN)	16 (9MFN, 7NZ)	12 (NZ)
6	0	3 (MFN)	1 (NZ)	5 (1 NZ, 4 NZ)

^1^ Histopathological grade definitions are given in Table 1.

## Data Availability

The data presented in this study are available on request from the corresponding author. fThe data are not publicly available due to restrictions e.g., their containing information that could compromise the privacy of research participants.

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
