# Peer review of "Local and Systemic Inflammation in Finnish Dairy Cows with Digital Dermatitis"

_animals, 2024, doi:10.3390/ani14030461_

Round 1
Reviewer 1 Report
Comments and Suggestions for Authors
The authors present here a histopathological study on digital dermatitis (Mortellaro’s disease). They studied the histological grading at the same time than descriptive scoring of lesions (M score) and research of inflammatory cytokines and proteins.
They propose a new lesion ‘s grading for histopathologists, they estimate closer to the M scoring than the precedent one.
It is proposed to the authors to be assisted for English reviewing. I am not able to correct the form because I’m not native English.
Abstract:
Abstract is comprehensive but the histopathological severity is not defined (We understand in the text that it is presence of necrosis)
Introduction
L45: “to be” is repeated.
L60:Title of reference number 24 is missing
L 91: Are the 8 control cows included in the 151 cows (and 104 final sampled cows)?
Material and methods
2.1. Study herds : The context would be rapidly described at least to understand what are those 81 dairy cows’ herds, and why only 25 were studied here. The explanations are not clear, the chart is better. Can the authors refer to the chart and explain why are 81 herds randomly chosen?
L90-92 : Can the authors specify, within the 151 cows, how many cows with lesions and cows without lesion? They have to put this summary at the same time than they specify the distribution per breed.
L109-113 and L 168-169: Did you do a bandaging or any protection of the 1.2 mm lesion done onto the foot? Did each biopsy punch pack individually, and individually identified? How did you identify the samples?
Line 117: Why did you put the biopsy into formalin (how many ml per biopsy?) for frozen it after? Or did the freezing only consider the RNAlater solution? It is to specify. What was the interest to put a biopsy sample into RNAlater solution? Dud you realized RNA analyses in another study? Why didn’t you present the results here? This would give possibly more interest for your study.
Table 1: The grades 2 and 3 have the same classification as “moderate hyperplasia” but the thickness is greater for grade 3 than 2. Are you sure not to have ulceration extending to the nonpapillar dermis on hyperplasia of 1.5 mm? In other terms the grades 2 and 3 are differentiated rather by the extension of ulceration or rather by the thickness?
Results
L211-212: Can you specify which are the healthy cows? Exclusively the 8 necropsic ones or 8+35M0?
L212-213 and L222: It is preferable to speak about tendency when statistics are not significative.
L234-238 : the sentence gives the same information than the just before one (severity has been defined as development of necrosis). It seems not necessary. I think that those informations would be better placed after the histological description of lesions (in paragraph 3.2).
L252-253: Is the description done here present in one of either figures? If yes, can you specify and indicate onto the figures?
L261-264: the sentence is too long and consecutively not clear. Can you improve it?
In general, in this paragraph can you refer to the figures, and give the figures more easy to read with arrows or circling of specific zones?
Figure 3-3b: can you put a bar to see the epidermis thickening?
Figures 4 to 5b: can you put an arrow to indicate the spirochetes, and also indicate the necrosis focus (all the readers are not histopathologists)?
Figure 6: can you mark the naked dermal papillae?
Discussion
L309-310: Can you explain or develop the idea : what is the importance to know how long the lesions exist in relation to SAA, IL-6 or Hp levels?
L316-321: Can you specify those assertions are generalized and not specific from cows?
L327-328: Are you sure that the world “indicate” is appropriate for this sentence. It’s only hypothesis not direct relation.
L342-343 : Can you refer to a publication for this sentence, otherwise it seems to be partial repetition of precedent information?
L351-356: Here you speak about weight-bearing surfaces. For my comprehension, the weight bearing surfaces are the sole and, the distal part of the hoof wall. Those are not usual tissues affected by the mortellaro’s disease and the description is more related to pathogenesis of sole ulcers. Why? If my comprehension is good, I don’t see the evidential relation with DD lesions.
L361-62: the spirochetes appear linked to necrosis. “secondary to” is not evidential, only a hypothesis. I agree with the authors but the form is too indicative. Similarly, L363 “is able to cause”… More over in the general plan of your discussion, why did not you join this with the discussion of the histopathological grades (L379-382)? In particular because your discussion L373 to 379 and 382-384 are not related with the precedent findings. The idea, if I understand well, is that the treponemas are around the animals and develop only when the skin present lesions, so it would not be an “imported disease”. If those points are interesting, the histopathological lesions alone cannot substitute to epidemiological one, specially with reference to molecular characterization of involved strains. So, your findings are not sufficient to discuss this point.
Chapter on pain is too long and as you have not reported the lameness score of the sampled cows, so you don’t avoid a broadline general discussion as those assertions are not substantiated by results. You only have evidence of IL-1 beta chemokine implication and not TNF nor IL-6. Only the final sentence (L407-409) is allowed by histological results, but it would be better if you have another reference than a PhD.
Conclusion : I suggest to avoid the reference to mechanical pressure as you have not any result illustrating this. I suggest also to speak about the level of TNF-alpha which is surprisingly more elevated in non-active lesion than in active one’s. In place of “created” I suggest to employ “propose”. Thereafter, in your table presence of spirochetes are not part of your new-grading. So can you change the sentence, which seems unclear (L418-420). Development would be more interesting in finding what are the causes leading to necrosis of skin (acidosis and acidity of feces, humidity, traumas, etc)
Ref 34 : Adult and fetal wound healing is the title!
Author Response
Reviewer 1
The authors present here a histopathological study on digital dermatitis (Mortellaro’s disease). They studied the histological grading at the same time than descriptive scoring of lesions (M score) and research of inflammatory cytokines and proteins.
They propose a new lesion ‘s grading for histopathologists, they estimate closer to the M scoring than the precedent one.
It is proposed to the authors to be assisted for English reviewing. I am not able to correct the form because I’m not native English.
Thank you for your review! We have now addressed the concerns You have raised, and we believe the manuscript has improved. Our answers are in blue.
Abstract:
Abstract is comprehensive but the histopathological severity is not defined (We understand in the text that it is presence of necrosis).
Thank You for noting this, we have now added a sentence: “A novel histopathological grading was developed based on the chronicity of the lesions and presence of necrosis and ulceration. The presence and number of spirochetes was graded separately.”. (l. 26-28)
Introduction
L45: “to be” is repeated.
We have corrected this. (l. 48)
L60: Title of reference number 24 is missing.
We have added the title. (l. 552)
L 91: Are the 8 control cows included in the 151 cows (and 104 final sampled cows)?
We apologize for writing the wrong number of control animals, the right number is 6. These animals are not included within in the 151 and 104 cows. We have made the flow chart (Fig. 1) better and referred to this chart in the text. (l.98)
We also added information to the manuscript text: “For comparison of epidermal thickness, biopsies were also taken from six cows originating from six different herds. These cows were euthanized in the Production Animal Hospital of the University of Helsinki due to non-lameness related reasons and samples were taken postmortem.” (l. 92-95)
Material and methods
2.1. Study herds: The context would be rapidly described at least to understand what are those 81 dairy cows’ herds, and why only 25 were studied here. The explanations are not clear, the chart is better. Can the authors refer to the chart and explain why are 81 herds randomly chosen?
Thank you for pointing this out. We have explained the process of choosing the herds in another study (Pirkkalainen and Riihimäki et al., 2021) and this is why we chose not to explain this in a detailed manner in this article. In the sentence: “For sampling, 25 farms were chosen based on the availability of a trimming chute or a professional hoof trimmer” (l. 80-81) we have explained the reason why only certain herds were chosen for biopsy sampling. We had a limited time for visiting the herds for our prevalence study (Pirkkalainen and Riihimäki et al., 2021) and sampling.
L90-92 : Can the authors specify, within the 151 cows, how many cows with lesions and cows without lesion? They have to put this summary at the same time than they specify the distribution per breed.
We have considered this, but we have shown the nr of lesions in Table 2 and believe this information is better to be placed in the Results section (l. 265-267).
L109-113 and L 168-169: Did you do a bandaging, or any protection of the 1.2 mm lesion done onto the foot? Did each biopsy punch pack individually, and individually identified? How did you identify the samples?
We used the same biopsy punch for both samples and started always with the RNA sample (l. 122-123). We added the requested information about bandaging (l. 128-134). We filled a form about the animal-related information (herd, EU-ear tag, breed, birth date etc.) and biopsy-related information (foot, macroscopic diagnosis, biopsy site etc.) and marked the sample vials with identification information (herd, animal, foot) (l. 133-134).
Line 117: Why did you put the biopsy into formalin (how many ml per biopsy?) for frozen it after? Or did the freezing only consider the RNAlater solution? It is to specify. What was the interest to put a biopsy sample into RNAlater solution? Dud you realized RNA analyses in another study? Why didn’t you present the results here? This would give possibly more interest for your study.
A very good observation! We did not freeze the formalin sample but kept it in room temperature. The RNAlater sample was used in another bacteriological study of our study group. However, there will be a separate article concerning these findings. We have added a sentence to specify the use of formalin (l. 125; l. 138-139).
Table 1: The grades 2 and 3 have the same classification as “moderate hyperplasia” but the thickness is greater for grade 3 than 2. Are you sure not to have ulceration extending to the nonpapillar dermis on hyperplasia of 1.5 mm? In other terms the grades 2 and 3 are differentiated rather by the extension of ulceration or rather by the thickness?
Thank You for bringing this up. The severity of lesions is defined by the thickness of lesions and presence of necrosis and suppurative inflammation. We have added a more precise description of grades 2 and 3 to Table 1 (l. 193-195) and made the table easier to read. The grade 3 is described as: “high moderate hyperplasia and ortokeratotic hyperkeratosis with necrosis and ulceration extending to the nonpapillar dermis.”
Results
L211-212: Can you specify which are the healthy cows? Exclusively the 8 necropsic ones or 8+35M0?
We have clarified the information about study animals and additional control animals for histopathology. We took additional biopsies from six cows and there are no serum samples available from these animals. Therefore, the healthy cows in Results are purely the 35 study cows with M0 diagnosis (l. 235). The six additional cows were looked at based on the age of cows and thickness of skin (l. 273-275).
L212-213 and L222: It is preferable to speak about tendency when statistics are not significative.
You are correct and we have changed the word “borderline significance” to “tendency” in lines 224 and 237.
L234-238 : the sentence gives the same information than the just before one (severity has been defined as development of necrosis). It seems not necessary. I think that those informations would be better placed after the histological description of lesions (in paragraph 3.2).
We thank you for Your comment. We categorized the lesions in many different ways, and this is also the reason why we have presented the results in multiple ways. We wanted to present these findings under the paragraph “3.1. Systemic and local inflammatory response to DD lesions” and therefore wrote the histological severity within the brackets. We added a sentence “Histopathological grading is explained in Table 1” in lines 262-263.
L252-253: Is the description done here present in one of either figures? If yes, can you specify and indicate onto the figures?
Yes, thank You for pointing this out. A naked dermal papilla is shown in Figure 3b and we have now referred o that Figure in the text (l. 278-279).
L261-264: the sentence is too long and consecutively not clear. Can you improve it?
We have now changed the sentence to make it more understandable. (l. 287-290)
In general, in this paragraph can you refer to the figures, and give the figures more easy to read with arrows or circling of specific zones?
Thank You for bringing this up. We have added labels, arrows and bars into the pictures.
Figure 3-3b: can you put a bar to see the epidermis thickening?
Yes, we have done this.
Figures 4 to 5b: can you put an arrow to indicate the spirochetes, and also indicate the necrosis focus (all the readers are not histopathologists)?
Yes, we have done this.
Figure 6: can you mark the naked dermal papillae?
Yes, we have done this.
Discussion
L309-310: Can you explain or develop the idea: what is the importance to know how long the lesions exist in relation to SAA, IL-6 or Hp levels?
The sentence “It is noteworthy that we collected blood samples only once, and we do not know how long different lesions had existed” is meant to illustrate the lack of information about the DD dynamics in this study. You have however raised an interesting question and we do believe the markers of inflammation to be elevated as long as the inflammation is present. (l. 345-346)
L316-321: Can you specify those assertions are generalized and not specific from cows?
We have added “in general” to the sentence referred. (l. 352)
L327-328: Are you sure that the world “indicate” is appropriate for this sentence. It’s only hypothesis not direct relation.
We change the sentence accordingly: “We could hypothesize that the bovine body is not able to eliminate DD from the skin because there is no efficient systemic reaction.” (l. 363)
L342-343 : Can you refer to a publication for this sentence, otherwise it seems to be partial repetition of precedent information?
We agree that the sentence was unclear. We have added the reference into the referred sentence. (l. 371-373)
L351-356: Here you speak about weight-bearing surfaces. For my comprehension, the weight bearing surfaces are the sole and, the distal part of the hoof wall. Those are not usual tissues affected by the mortellaro’s disease and the description is more related to pathogenesis of sole ulcers. Why? If my comprehension is good, I don’t see the evidential relation with DD lesions.
We have adjusted the referred sentence: “The necrosis of the dermal papillae appears to precede the necrosis of the epidermis and it is probably ischemic. The thickened epidermis causes pressure and shear stress on the delicate dermal papillae, which impairs the blood flow and causes hypoxia, leading to ischemic necrosis followed by erosion and ulceration.” (l. 387-390). We believe it is now easier to understand that the pressure we refer to is due to thickening of the skin, not biomechanical weight bearing due to ground contact. However, this skin area is under strain while the cow is moving and standing even though it not the weight bearing surface of cows’ weight.
L361-62: the spirochetes appear linked to necrosis. “secondary to” is not evidential, only a hypothesis. I agree with the authors but the form is too indicative. Similarly, L363 “is able to cause”… More over in the general plan of your discussion, why did not you join this with the discussion of the histopathological grades (L379-382)? In particular because your discussion L373 to 379 and 382-384 are not related with the precedent findings. The idea, if I understand well, is that the treponemas are around the animals and develop only when the skin present lesions, so it would not be an “imported disease”. If those points are interesting, the histopathological lesions alone cannot substitute to epidemiological one, specially with reference to molecular characterization of involved strains. So, your findings are not sufficient to discuss this point.
Chapter on pain is too long and as you have not reported the lameness score of the sampled cows, so you don’t avoid a broadline general discussion as those assertions are not substantiated by results. You only have evidence of IL-1 beta chemokine implication and not TNF nor IL-6. Only the final sentence (L407-409) is allowed by histological results, but it would be better if you have another reference than a PhD.
We changed the sentence where we used earlier the words “secondary to”: “According on our findings, the spirochetes appear to be present only in the necrotic skin. Thus, we hypothesize that they are secondary invaders.”. (l. 396-397). We know that some of the Treponemes are more virulent and seem to be related to more severe DD-lesions. However, as we stated, most of the lesions we found were quite mild. Yet our histological findings suggest a strong environmental effect. As we still do not see severe or to be more precise active DD-lesions in most Finnish herds, we might be lacking the more virulent Treponema strains. There is an ongoing study by our study group which will shed light to this aspect in the near future. Until then, we can merely hypothesize on this matter in this article. We added a sentence to sum this idea up: “As animal trade has been identified as a risk factor for DD [44], it can be hypothesized that we might not have imported that many virulent Treponemes to Finland.” (l. 419-421)
We believe that the chapter concerning pain is justifiable. We did not find elevated APPs and thus the systemic inflammatory reaction does not seem to be present. However, we wanted to understand why DD has systemic effects such as loss of milk yield. This is the reason to discuss the role of pain – a very important welfare aspect.
There are many articles tackling the problem of dry heels in humans. We have added another reference and explained more thoroughly the relationship between epidermal thickening, cracking and necrosis and pain. (l. 445-449)
Conclusion: I suggest to avoid the reference to mechanical pressure as you have not any result illustrating this. I suggest also to speak about the level of TNF-alpha which is surprisingly more elevated in non-active lesion than in active one’s. In place of “created” I suggest to employ “propose”. Thereafter, in your table presence of spirochetes are not part of your new-grading. So can you change the sentence, which seems unclear (L418-420). Development would be more interesting in finding what are the causes leading to necrosis of skin (acidosis and acidity of feces, humidity, traumas, etc).
We have adjusted the sentence (l.474-476). This type of pressure does not come from weight bearing, but thickening of the skin. We believe we have now illustrated the idea better. We have changed the word “created” into “propose” (l. 474). Our histopathological grading is based on the age and severity of the lesions and presence of spirochetes.
TNFα is known to be elevated both in acute and chronic inflammation, so it is not a surprise that is was elevated in chronic DD-lesions. And we must remember, that the macroscopical diagnosis did not always find the active lesions compared to histopathological grading.
Ref 34 : Adult and fetal wound healing is the title!
Thank You for noticing, we have corrected this. (l. 576)
Reviewer 2 Report
Comments and Suggestions for Authors
animals-2776524
Local and systemic inflammation in Finnish dairy cows with 2 digital dermatitis
REMARKS AND RECOMMENDATIONS
This is an interesting paper describing the local and systemic inflammation in Finnish dairy cows with digital dermatitis.
I made several comments and questions in the report below (and in the pdf - see the yellow markings); please answer these questions and comments and incorporate all these arguments in your revised manuscript.
Line 29: Spirochetes were found only in samples from necrotic lesions; You should add here … in M2 and M4.1 lesions.
Line 50: You should cite also newer studies on histopathology of DD such as Alsaaod M, Jensen TK, Miglinci L, Gurtner C, Brandt S, Plüss J, et al. (2022): Proof of an optimized salicylic acid paste-based treatment concept of ulcerative M2-stage digital dermatitis lesions in 21 dairy cows. PLoS ONE 17(6):e0269521. https://doi.org/10.1371/journal.pone.0269521.
Line 78: In the Materials & Methods section you should add some information, whether you did check also the general health status of all these selected cattle by performing a clinical exam such as pulse rate, respiratory rate, rectal temperature, presence of metabolic disorders, signs of subclinical and clinical mastitis etc., and for presence of other painful claw lesions which might have an (much more important) impact on the systemic inflammatory response. Please add this crucial information. If the general health status of all these selected cattle was not checked, you should discuss this fact as a crucial limiting aspect of your study.
Line 268-272, table 3: I propose to add here in this table also the type of DD stage from which the biopsy samples were taken for histopathology.
Line 273-274: Your results regarding the detection of Treponema in biopsy samples of the different M-stages is in contrast to the results of Capion et al 2012: Infection dynamics of digital dermatitis in first-lactation Holstein cows in an infected herd. J. Dairy Sci. 95 :6457–6464; they found Treponemes in M1, M2, M3, M4 and M4.1 stages. Please discuss this fact.
Line 282, Fig. 3: I propose to add here in this figure also the type of DD stage from which this histopathologic section/staining was made.
Line 286, Fig. 4ab: I propose to add here in this figure also the type of DD stage from which this histopathologic section/staining was made.
Line 290, Fig. 5a,b: I propose to add here in this figure also the type of DD stage from which this histopathologic section/staining was made.
Line 294, Fig. 6: I propose to add here in this figure also the type of DD stage from which this histopathologic section/staining was made.
Line 297, Fig. 7: I propose to add here in this figure also the type of DD stage from which this histopathologic section/staining was made.
Line 328: Krull et al. [35] stated that it took approximately over four months for a normal skin to develop into an advanced DD lesion, with …: Sorry, your statement here is not correct, because KRULL et al. reported that it took at mean 133 days for development of a clinical M stage, but the median was 105 d, and the minimum “incubation period” was only 38 days. Also, Holzhauer et al (2008): Clinical course of digital dermatitis lesions in an endemically infected herd without preventive herd strategies. Vet J 177, 222-230 reported much shorter periods for development of M2 lesions. Further, the time period for development of active DD lesions may also depend on the fact that a DD-free herd is infected by Treponemes at the first time and by individual cow-related factors such a cow-related genetical susceptibility (see Capion et al. 2012; Biemans et al. 2019). Please rephrase your sentence, and I suggest using the median instead of the mean value, and also the range (min – max).
Line 368: Here you should also compare the results of this recently published study with your histopathological findings: Alsaaod M et al. (2022) Proof of an optimized salicylic acid paste-based treatment concept of ulcerative M2-stage digital dermatitis lesions in 21 dairy cows. PLoS ONE 17(6):e0269521. https://doi.org/10.1371/journal.pone.0269521.
Line 412: A section on limitations of your study is missing; It should be mentioned that the total number of cattle examined was low with only 104 animals and that there were only 7 - 35 DD lesions per M-stage. Furthermore, it is not known to which type of genetic susceptibility to DD the examined cows belonged (type 2 or 3 according to Capion et al. 2012); this aspect may also have an influence on the concentrations of pro-inflammatory cytokines in the blood. It must also be mentioned that it may not be possible to rule out that other metabolic diseases or mastitis etc. that may be present in the examined cows (sub)clinically were mainly or partly responsible for the measured concentrations of pro-inflammatory cytokines. These crucial aspects concerning the assignment of the measured cytokines to possible unknown (?) but simultaneously occurring inflammatory processes in the animal body should be referred in the discussion.
Line 422: Here you should add a sentence presenting the clinical relevance of your results for veterinarians: e.g., can the assessment of proinflammatory cytokines in the blood of cattle with uncertain clinical DD status be helpful for clarification of their definitive DD status (e.g. in the case of a planned purchase of cows), and would it be possible to differentiate different pathologies (DD, metabolic disorders, mastitis ...) by assessment of the type and concentration of proinflammatory cytokines?
Author Response
Reviewer 2
This is an interesting paper describing the local and systemic inflammation in Finnish dairy cows with digital dermatitis.
I made several comments and questions in the report below (and in the pdf - see the yellow markings); please answer these questions and comments and incorporate all these arguments in your revised manuscript.
Thank you for Your insightful review! We have now adjusted the manuscript and believe it is improved. We agree on the lack of macroscopic DD-diagnosis and have added these and labelling to the Figures. Please find our answers in blue.
Line 29: Spirochetes were found only in samples from necrotic lesions; You should add here … in M2 and M4.1 lesions.
Thank you for your comment. However, we found spirochetes in other macroscopic diagnoses. For example, some of the M4 and even M0 lesions had necrosis and spirochetes present. This is why we chose to present the result with histopathological grading. Therefore, we cannot add the suggested part into the text. (l. 30-31)
Line 50: You should cite also newer studies on histopathology of DD such as Alsaaod M, Jensen TK, Miglinci L, Gurtner C, Brandt S, Plüss J, et al. (2022): Proof of an optimized salicylic acid paste-based treatment concept of ulcerative M2-stage digital dermatitis lesions in 21 dairy cows. PLoS ONE 17(6):e0269521. https://doi.org/10.1371/journal.pone.0269521.
An excellent suggestion and an interesting article. However, the article suggested focuses more on the effect of treatment on DD. The description of histopathological changes is similar to the other articles referred, and has been evaluated only from M2 and healed M5 lesions. We have added a line in the discussion referring to the article by Alsaood et al. (l. 404)
Line 78: In the Materials & Methods section you should add some information, whether you did check also the general health status of all these selected cattle by performing a clinical exam such as pulse rate, respiratory rate, rectal temperature, presence of metabolic disorders, signs of subclinical and clinical mastitis etc., and for presence of other painful claw lesions which might have an (much more important) impact on the systemic inflammatory response. Please add this crucial information. If the general health status of all these selected cattle was not checked, you should discuss this fact as a crucial limiting aspect of your study.
Thank you for pointing this out. We added a more detailed description concerning the health status of the cow: “A brief clinical examination was performed to exclude other diseases. We measured the body temperature and observed whether the animal had ocular or nasal discharge or abnormal respiratory rate. The farmer was asked whether the cow had had any signs of mastitis, lameness, or other diseases prior to sampling. Also, records concerning hoof trimming were evaluated. Cows with concurrent pathologies were not taken into the study.”. (l. 112-117)
We also added a few lines of what type of information was gathered during the sampling (l. 132-133). As we also received the milk recording information, one cow was detected to have a mastitis and was eliminated from the study later (Figure 1).
Line 268-272, table 3: I propose to add here in this table also the type of DD stage from which the biopsy samples were taken for histopathology.
Thank you for your suggestion. We thought about this, but it would make the table harder to understand. As we saw that the novel histopathological grading is more precise in grading the lesions, we prefer to use it in Table 3.
Line 273-274: Your results regarding the detection of Treponema in biopsy samples of the different M-stages is in contrast to the results of Capion et al 2012: Infection dynamics of digital dermatitis in first-lactation Holstein cows in an infected herd. J. Dairy Sci. 95 :6457–6464; they found Treponemes in M1, M2, M3, M4 and M4.1 stages. Please discuss this fact.
We found treponemes in all M-lesion categories (including M0) and therefore do not believe our findings are in contrast to the findings of Capion et. al (2012). This is also why we chose to elaborate the novel histopathological grading. At farm visits we are not able to detect the lesion severity always merely based on macroscopical findings. (Table 3).
Line 282, Fig. 3: I propose to add here in this figure also the type of DD stage from which this histopathologic section/staining was made.
Thank you for the suggestion, we have added this information.
Line 286, Fig. 4ab: I propose to add here in this figure also the type of DD stage from which this histopathologic section/staining was made.
Thank you for the suggestion, we have added this information.
Line 290, Fig. 5a,b: I propose to add here in this figure also the type of DD stage from which this histopathologic section/staining was made.
Thank you for the suggestion, we have added this information.
Line 294, Fig. 6: I propose to add here in this figure also the type of DD stage from which this histopathologic section/staining was made.
Thank you for the suggestion, we have added this information.
Line 297, Fig. 7: I propose to add here in this figure also the type of DD stage from which this histopathologic section/staining was made.
Thank you for the suggestion, we have added this information.
Line 328: Krull et al. [35] stated that it took approximately over four months for a normal skin to develop into an advanced DD lesion, with …: Sorry, your statement here is not correct, because KRULL et al. reported that it took at mean 133 days for development of a clinical M stage, but the median was 105 d, and the minimum “incubation period” was only 38 days. Also, Holzhauer et al (2008): Clinical course of digital dermatitis lesions in an endemically infected herd without preventive herd strategies. Vet J 177, 222-230 reported much shorter periods for development of M2 lesions. Further, the time period for development of active DD lesions may also depend on the fact that a DD-free herd is infected by Treponemes at the first time and by individual cow-related factors such a cow-related genetical susceptibility (see Capion et al. 2012; Biemans et al. 2019). Please rephrase your sentence, and I suggest using the median instead of the mean value, and also the range (min – max).
Thank you for noticing our error, we have now corrected the sentence (l. 364-365).
Line 368: Here you should also compare the results of this recently published study with your histopathological findings: Alsaaod M et al. (2022) Proof of an optimized salicylic acid paste-based treatment concept of ulcerative M2-stage digital dermatitis lesions in 21 dairy cows. PLoS ONE 17(6):e0269521. https://doi.org/10.1371/journal.pone.0269521.
Thank you for the suggestion. We have now referred also to Alsaood et al. (l. 404). However, they have evaluated biopsies only from M2 and M5 (healed) lesions. We did find hyperplasia and necrosis from all M2 lesions and Alsaood et al. (2022) described the M2 lesions with ulcerations and moderate to severe perivascular, chronic, lymphoplasmacytic dermatitis. We did not evaluate healed lesions and cannot therefore compare the findings in that aspect.
Line 412: A section on limitations of your study is missing; It should be mentioned that the total number of cattle examined was low with only 104 animals and that there were only 7 - 35 DD lesions per M-stage. Furthermore, it is not known to which type of genetic susceptibility to DD the examined cows belonged (type 2 or 3 according to Capion et al. 2012); this aspect may also have an influence on the concentrations of pro-inflammatory cytokines in the blood. It must also be mentioned that it may not be possible to rule out that other metabolic diseases or mastitis etc. that may be present in the examined cows (sub)clinically were mainly or partly responsible for the measured concentrations of pro-inflammatory cytokines. These crucial aspects concerning the assignment of the measured cytokines to possible unknown (?) but simultaneously occurring inflammatory processes in the animal body should be referred in the discussion.
You are correct. We have now created a new chapter explaining the study limitations (l. 452-463).
Line 422: Here you should add a sentence presenting the clinical relevance of your results for veterinarians: e.g., can the assessment of proinflammatory cytokines in the blood of cattle with uncertain clinical DD status be helpful for clarification of their definitive DD status (e.g. in the case of a planned purchase of cows), and would it be possible to differentiate different pathologies (DD, metabolic disorders, mastitis ...) by assessment of the type and concentration of proinflammatory cytokines?
Thank You for Your suggestion. However, the aim of our study was not to evaluate the applicability of cytokine and APP diagnostics for clinical use. We wanted to understand the local and systemic reactions created by DD. We do not believe that cytokine and APP diagnostics are applicable for practical clinical use.
Reviewer 3 Report
Comments and Suggestions for Authors
Overall, this article discusses the findings quite thoroughly. The problem statement and objectives were quite clear.
Methodology - Some parts of the methodology are not described (as mentioned in the comment sections). Se and SP of the test kits to be included as well.
Results - Overall writing can be improved especially where a lot of values are presented in the text - perhaps to be tabulated. Some of the figures and Tables need more and proper labeling to ease the readers to understand what they are looking at.

Comments on the Quality of English Languagewell-written article but could be improved in certain areas to ease reading for the audience
Author Response
Reviewer 3
Overall, this article discusses the findings quite thoroughly. The problem statement and objectives were quite clear.
Methodology - Some parts of the methodology are not described (as mentioned in the comment sections). Se and SP of the test kits to be included as well.
Results - Overall writing can be improved especially where a lot of values are presented in the text - perhaps to be tabulated. Some of the figures and Tables need more and proper labeling to ease the readers to understand what they are looking at.
Thank you for your comments! We have now adjusted the manuscript and believe it is improved. We agree on the lack of labelling and explanatory arrows and have changed the Figures accordingly. Please find our answers in blue.
Again, the numbers 104 do not tally with 151 in line 85. This sentences is best to be placed later after the explanation for Figure 1.
Thank you for pointing this out. We have improved the sentence and wish to keep it under the study population chapter. (l. 96-98)
Was this conducted as blinded or not to elaborate more on the assessment.
Yes, they were blinded. We have added a sentence on lines 100-101: “The other two researchers did not have any information of the cow or lesions prior to evaluation.”
What does this sentence means? other legs are affected as well with DD?
It means that we made a macroscopic diagnosis on the other foot also. The diagnoses ranged from M0-M4.1 and as said in the text, the most severe diagnosis (foot) wash chosen for sampling. The presence of another DD-diagnosis was taken into account in statistical analysis. (l. 109-111)
Observation was conducted but not clinical examination.
Thank You for Your comment. We did a brief clinical examination. We added information about this in lines 112-117.
How was the animal restraint at this point?
The cows were taken into a trimming chute and the foot with the most severe lesion was lifted up. We have added a sentence concerning this aspect. (l. 118)
To state the Se and SP of each test use
As we did not use the cytokine or APP test kits for any kind of diagnosis, we believe Se and Sp are not necessary information and not available. There are no cut off values available for this type of test kits and laboratories are recommended to establish their own reference range.
How was it confirmed as spirochetes?
In our histopathological investigations we used Warthin-Starry silver nitrate-based staining to detect spirochetes. As this is a commonly used method, we did not find it necessary to explain the method in detail. (l. 183-184)
We understand that also other methods like immunohistochemical methods, to detect spirochetes. We have yet unpublished material where we used PCR, FISH and microbiome investigations to detect Treponemes.
How was this conducted, please elaborate.
Please find our answer in the previous question.
Is this based on any figure or table? can this be tabulated to ease understanding?
Thank you for bringing this up. We did not make a table for cytokine and APP comparisons with different histopathological grades. We believe the text is sufficient, as only interleukin-1β was elevated.
How was this confirmed? not stated in methodology at all. please elaborate (Table 3).
In our histopathological investigations we used Warthin-Starry silver nitrate-based staining to detect spirochetes. As this is a commonly used method, we did not find it necessary to explain the method in a more precise way. We added some information to make the evaluation of presence of spirochetes more understandable to the text (l. 183-185) and Table 3.
How is this define as? low, moderate or large - should be defined and explained in the methodology section. (Table 3)
The grading is semiquantitative and it is written in MM section in lines 183-185. The grading and explanation are commonly used in pathology, as it is difficult to visualize the exact number of spirochetes by microscopy.
Figure needs proper labelling and arrows.
We have now added labelling and arrows.
Please indicate in the figure “thickness by hyperplasia”.
We have added a bar to better evaluate the thickness of epidermis. (Figures 3a-b)
Spirochetes à please indicate with arrow
We have now added arrows indicating spirochetes. (Figures 4b, 5b)
Figures: 5a and 5b à needs proper labelling as to which is a n b.
We have now added labelling.
Naked dermal papillae à to include labelling for better understanding of the histo. similar comment for other figures as well
We have now added labelling and arrows. (all Figures)
Round 2
Reviewer 2 Report
Comments and Suggestions for Authors
REMARKS AND RECOMMENDATIONS
The authors have incorporated all my comments and questions into the revised version of the manuscript.
For obvious clinical reasons, I suggest that you place lines 112-116 (“A brief clinical examination was performed to exclude other diseases. We measured the body temperature and observed whether the animal had ocular or nasal discharge or abnormal respiratory rate. The farmer was asked whether the cow had had any signs of mastitis, lameness, or other diseases prior to sampling. Also, records concerning hoof trimming were evaluated. Cows with concurrent pathologies were not taken into the study.”) at the beginning of section 2.2 on line 86. Then, you can continue with “The digital lesions on …).
Author Response
Reviewer 2
The authors have incorporated all my comments and questions into the revised version of the manuscript.
For obvious clinical reasons, I suggest that you place lines 112-116 (“A brief clinical examination was performed to exclude other diseases. We measured the body temperature and observed whether the animal had ocular or nasal discharge or abnormal respiratory rate. The farmer was asked whether the cow had had any signs of mastitis, lameness, or other diseases prior to sampling. Also, records concerning hoof trimming were evaluated. Cows with concurrent pathologies were not taken into the study.”) at the beginning of section 2.2 on line 86. Then, you can continue with “The digital lesions on …).
We are thankful for the review and suggestion. We have now relocated the paragraph concerning clinical examination and it begins now from line 93.